# Lymph Nodes as Anti-Tumor Immunotherapeutic Tools: Intranodal-Tumor-Specific Antigen-Pulsed Dendritic Cell Vaccine Immunotherapy

**DOI:** 10.3390/cancers14102438

**Published:** 2022-05-15

**Authors:** Takashi Morisaki, Takafumi Morisaki, Makoto Kubo, Shinji Morisaki, Yusuke Nakamura, Hideya Onishi

**Affiliations:** 1Fukuoka General Cancer Clinic, Fukuoka 812-0018, Japan; shinji.m.03235@gmail.com; 2Department of Surgery and Oncology, Graduate School of Medical Sciences, Kyushu University, Fukuoka 812-8582, Japan; moritaka@med.kyushu-u.ac.jp (T.M.); kubo.makoto.804@m.kyushu-u.ac.jp (M.K.); 3Department of Cancer Therapy and Research, Graduate School of Medical Sciences, Kyushu University; Fukuoka 812-8582, Japan; ohnishi.hideya.928@m.kyushu-u.ac.jp; 4Department of Medicine and Clinical Science, Graduate School of Medical Sciences, Kyushu University, Fukuoka 812-8582, Japan; 5Cancer Precision Medicine Center, Japanese Foundation for Cancer Research, Tokyo 135-8550, Japan; yusuke.nakamura@jfcr.or.jp

**Keywords:** lymph nodes, dendritic cells, stromal cells, intranodal, cancer vaccines, neoantigen, peptides

## Abstract

**Simple Summary:**

In the field of cancer therapy, lymph nodes are important not only as targets for metastases resection but also as prudent target organs for cancer immunotherapy. Lymph nodes comprise a complete structure for the accumulation of a large number of T cells and their distribution throughout the body after antigen presentation and activation of dendritic cells. This review highlights current topics on the importance of lymph node structure in antitumor immunotherapy and intranodal-antigen-presenting mature dendritic cell vaccine therapy. We also discuss the rationale behind intranodal injection methods and their applications in neoantigen vaccine therapy, a new cancer immunotherapy.

**Abstract:**

Hundreds of lymph nodes (LNs) are scattered throughout the body. Although each LN is small, it represents a complete immune organ that contains almost all types of immunocompetent and stromal cells functioning as scaffolds. In this review, we highlight the importance of LNs in cancer immunotherapy. First, we review recent reports on structural and functional properties of LNs as sites for antitumor immunity and discuss their therapeutic utility in tumor immunotherapy. Second, we discuss the rationale and background of ultrasound (US)-guided intranodal injection methods. In addition, we review intranodal administration therapy of tumor-specific-antigen-pulsed matured dendritic cells (DCs), including neoantigen-pulsed vaccines.

## 1. Introduction

The most frequent organs of cancer metastasis are LNs. In addition, resection of LNs near tumors is one of the most important therapeutic measures in cancer treatment [1,2,3]. Thus, combined resection of regional LNs with the tumor assists in preventing distant metastasis and local recurrence of tumors. This measure is also an essential tool for predicting prognosis and determining the need for adjuvant therapy through precise cancer staging [4,5,6]. Moreover, the immune microenvironment in tumor-neighboring LNs, such as tumor-draining LNs (TDLNs), is known to be immunosuppressive, which favors the migration of cancer cells into LNs in which they proliferate [7].

On the other hand, the importance of LNs as sites for inducing immune responses against tumors has been reconfirmed: they are the first sites where antigen-presenting cells, such as DCs that have taken up tumor antigens in tumor proximity, enter through the import of the lymphatic vessels to induce and amplify the immune response of tumor-antigen-responsive T cells [8,9]. It has been shown that lymphocyte infiltration in the tumor area and immune responsiveness of TDLN are important for the efficacy of immune checkpoint inhibitors (ICIs), which have caused paradigm shifts in cancer treatment [10]. Regional LNs are essential in antitumor immunity in MSI-high colorectal cancer and warned of excessive LN dissection [11].

Furthermore, LNs near tumors and normal ones in other areas are complete immune organs: both comprise immune and stromal cells. Immune cells in LNs include myeloid cells, such as DCs and macrophages, and lymphoid cells, such as T lymphocytes and B lymphocytes [8]. Stromal cells have been shown to play an important role in immune response initiated in LNs [12,13]. The primary function of LNs in tumor immunity includes encountering antigen-presenting cells with T lymphocytes and subsequent activation and proliferation of antigen-reactive T lymphocytes. In the first part of this review, the characteristics of LNs in tumor immunity as sites for initiation of antigen-specific immune responses are discussed, along with the latest findings on various cell types found in LNs.

To create cancer vaccines, it is important to consider what to use as the antigen, how to deliver that antigen to the body’s immune cells, and the route of administration of the vaccine [14]. One method of cancer vaccine therapy is to pulse cancer-specific antigens to patients’ dendritic cells outside the body, which are then administered into the LNs. This therapy is thought to efficiently stimulate and activate antigen-responsive T lymphocytes directly in LNs. The main purpose of the latter half of this paper is to review the reported therapies of intranodal injection of DC vaccines. We also focus on the theoretical rationale behind the LN injection technique, using the lymphangiography method of injecting oil-based contrast media into inguinal lymph nodes as an example. We also present the practicalities of LN injections as there have been no reports describing this technique in detail. Finally, we discuss the usefulness of the intranodal neoantigen peptide-pulse DC vaccine that we have recently initiated.

## 2. Structure of LNs and Their Constituent Cells

Normal LNs, numbering over 450–500, are small immune tissue structures scattered throughout the body [15]. They are important secondary lymphoid tissues connecting the innate and acquired immune systems. Although each LN is small in size, it can be considered a complete immune organ [8].

The superficial layer of the LN is covered with a capsule, and multiple afferent lymphatic vessels, which collect lymph fluids, DCs, and lymphocytes from surrounding tissues, flow into the node through the capsule [16]. The superficial layer of the LN contains lymphoid follicles composed of B lymphocytes and stromal cells, while the paracortical region contains a large number of T lymphocytes and DCs [8,9,15]. The medulla is at the center of the LN, in which T lymphocytes, DCs, and stromal cells (also called fibroblastic reticular cells (FRCs)) accumulate [8,9,12]. The advanced part of the medulla, called the hilus, contains arteries and veins that flow into LNs and efferent lymphatic vessels that carry T lymphocytes and dendritic cells from the medulla to the upstream lymph duct [8,9,15].

Various subsets of immune cells are contained in LNs, including antigen-presenting DCs, CD8-positive cytotoxic T cells, CD4-positive helper T cells, and B lymphocytes, which are transformed into plasma cells responsible for antibody production through antigen stimulation and activation by helper T cells [8,9,15]. DCs receive antigens in the perinodal tissues, reach LNs through the afferent lymphatic vessels, and circulate among numerous T lymphocytes to find T lymphocytes that respond to antigenic peptides, resulting in the activation and proliferation of antigen-responsive CD8 and CD4 T lymphocytes [16]. Conversely, T lymphocytes circulate among the stromal cells of LNs, seeking to encounter DCs that present antigens to which they respond [8,15]. T lymphocytes that have undergone antigen-specific activation migrate into the efferent lymphatic vessels and blood circulatory system. They spread throughout the body, respond directly or indirectly to antigens, and are involved in the elimination of other cells that carry pathogenic and tumor-associated antigens [8,9,15].

Recent studies have revealed that LN stromal cells, other than immune cells, are also actively involved in the accumulation of immune cells and regulation of immune responses, in addition to being involved in the LN structure itself [12,13]. For example, LNs, which are normally only a few millimeters in size, may swell to several centimeters in size when an acute immune response occurs in the surrounding tissues. The cells that carry the plasticity of these LNs are LN stromal cells (LNSCs), especially FRCs, which are stromal cells that are abundant in the paracortical regions and medulla [12,13]. Normally, FRCs express podoplanin (PDPN) and are in a contractile state. However, upon contacting mature DCs, CLEC-2 binding inhibits PDPN signaling, leading to relaxation of the actin-myosin system of these FRCs, which in turn leads to myosin formation. The myosin system of FRCs relaxes, and their cell diameter increases, resulting in the enlargement of the LN itself [13].

One of the important functions of the FRC is the formation of the conduit system, which is the microfiber of the extracellular matrix through which lymph fluid containing antigens and inflammatory mediators flows, contributing to the control of immune cells in the lymph nodes [17].

Another function of FRCs is to accumulate lymphocytes and DCs from surrounding tissues. For example, follicular DCs (FDCs) in cortical lymphoid follicles produce CXCL13 and accumulate B lymphocytes, bearing their ligand CXCR5. In the paracortical region, FRCs produce CCL19 and CCL21, which cause DCs and T lymphocytes bearing their CCR7 receptor to accumulate around themselves [12]. In addition, FRCs produce IL-7, a cytokine necessary for T lymphocytes’ survival and maintenance of their activity [18].

The influx of immune cells such as T cells into LNs occurs mainly via the high endothelial venule (HEV), which branches off from the LN arteries and veins. Additionally, HEV is a unique blood vessel found in LNs. Generally, HEVs produce the peripheral node addressin, which promotes the rolling of naïve and memory lymphocytes with L-selectin ligands into HEVs. Subsequently, LFA-1 on the surface of lymphocytes binds strongly to ICAM-I/II on the surface of HEVs [19]. The binding of lymphoid cells to HEVs is followed by their migration through HEVs and into LNs by HEV-produced CCL21 [19].

After receiving antigens from surrounding tissues, DCs enter LNs through the afferent lymphatic vessels and migrate to the paracortex [16]. Activated antigen-stimulated T cells flow out of LNs via the efferent lymphatic vessels into the blood circulation. It has been reported that mature DCs in LNs direct the proliferation of FRCs via lymphotoxin-β and promote vascular endothelial growth factor (VEGF) production from FRCs, which in turn transforms the vascular endothelium in LNs into HEVs [19]. Both HEVs and afferent and efferent lymphatic vessels are involved in immune cell migration pathways and the activation and regulation of immune cells.

Thus, LNSCs are cells acting as regulators of various types of immune cell functions in LNs and controlling changes in the three-dimensional microarchitecture in response to surrounding conditions such as inflammation and tumor [12,13,20]. Figure 1 shows a schematic diagram of a single LN.

## 3. Tumor Immune Microenvironment and Local Immunotherapy in Tumor Drainage LNs

As described above, LNs are thought to play an essential role as a site of immune induction in both infectious and tumor immunity. Recently, regarding tumor immunity, the tumor microenvironment (TME), which is the site of direct activation of immune cells against the tumor, has received much attention, and its immunological environment has been well understood [21]. However, T lymphocytes that react to tumor cells are stimulated and activated by antigens and proliferate only in nearby LNs. In particular, TDLNs are the sites where antigen-presenting cells that have taken up tumor antigens activate antigen-reactive naïve T lymphocytes. TDLNs are important for determining the extent of LN dissection, as well as sentinel nodes in surgical procedures for tumors such as breast cancer and melanoma [7].

Furthermore, TDNLs and other LNs in the tumor vicinity have been found to be in an immune microenvironment that promotes cancer cell metastasis and proliferation, even before cancer metastasizes [7,22,23]. For example, DCs, which induce tumor antigen presentation to and activation of T lymphocytes, are mostly in an immature state in tumor-area LNs and may induce immune tolerance to tumor-antigen-specific T cells [7,23,24]. It is also known that there are many regulatory T cells in TDLNS that reduce immune responses [7,23]. Furthermore, lymphatic endothelial cells are known to produce chemokines that promote the migration of tumor cells into LNs [22,23].

Therefore, most LNs in the tumor vicinity, including TDLNs, are highly immunosuppressive, providing an immune environment that induces a negative immune response to tumors. Consequently, various measures should be taken to improve this environment [24,25].

However, in a mouse tumor model, it has been recently shown that the removal of TDLNs before administration of ICIs abolished the antitumor effect of these ICIs [10,26]. They also have indicated that removal of TDLNs after administration of ICIs attenuated their effects, demonstrating that TDLNs play a very important role in enhancing the therapeutic efficacy of ICIs [10,26].

Since the importance of TDLNs in antitumor immunity has been demonstrated, methods to enhance the direct antitumor immune response in TDLNs have been attempted. Attempts to enhance the antitumor immune response in TDLNs have been initiated by injecting small amounts of ICIs and immune adjuvants, such as TLRs, into the tumor or around LNs near the tumor [27,28].

In addition, the development of vaccine and drug therapy using nanotechnology targeting LNs has recently been reported and is attracting attention as a method that can induce a reliable immune response in LNs while reducing systemic adverse events; moreover, CpG-DNA/peptide vaccine conjugated with lipophilic albumin to enhance vaccine accumulation in LNs is one of these inventions. The efficacy of the vaccine was confirmed in a mouse tumor model by demonstrating higher immune response and antitumor efficacy and lower systemic toxicity of the node-accumulating vaccine compared to that following systemic administration [29]. A multistage accumulation system of drugs in LNs and specific immune cells within LNs is another method currently under investigation [30]. Both papers reported the importance of the accumulation of vaccines or drugs in the lymph nodes.

## 4. DC Vaccine Therapy

Generally, DCs are potent immune amplifiers essential for the activation and proliferation of antigen-reactive T lymphocytes in cancer and viral infections. They have greater antigen-presenting ability and can present foreign antigens to both CD8- and CD4 T cells by cross-presentation. Therefore, they are expected to be utilized in cancer-specific vaccine therapy [31]. Although cancer immunotherapy using DCs has been in use for more than 20 years, it has a disadvantage: it is not a standard treatment method because it is different from drugs in nature. In other words, it is a cell therapy in which a patient’s DCs are equipped with this patient’s own cancer antigen information (cancer-specific antigen); thus, this can be called the ultimate tailor-made or individualized therapy.

The safety and efficacy of DC vaccines have been clarified in many clinical trials up to phases 1–2 [32]. In DC vaccine therapy, monocyte-derived DCs induced by GM-CSF+IL-4 from peripheral blood monocytes are usually used for treatment [31,32,33]. Recently, ICIs, which have caused paradigm shifts in cancer treatment, have already become the standard of care in several oncology fields. However, only a small percentage of patients benefit from ICIs due to problems such as immune-related adverse events and hyperprogression [34,35]. Although ICIs can restore T cell exhaustion, there are issues to be resolved, such as the priming and activation of T cells by tumor-antigen-presenting cells and their accumulation in tumors. It has been shown that DC vaccine therapy is one of the immunotherapies that can overcome such hurdles, and its importance has recently been reaffirmed [33].

## 5. The Rationale for Intranodal Administration of DCs and Intranodal Contrast Injection

The most common route of administration of DC vaccines is intradermal administration. However, the intranodal administration of cell vaccines has several unique features compared to the intradermal approach. As mentioned above, many T lymphocytes stay in the LNs, and there is a constant influx of T lymphocytes via HEVs; therefore, as long as antigen-presenting mature DCs are present, tumor antigen-reactive T lymphocytes are likely to encounter them, become activated, and proliferate. Hence, theoretically, direct injection of mature DCs presenting tumor antigens to LNs is more likely to activate antigen-responsive T lymphocytes, potentially inducing an efficient and rapid tumor-specific T cell response. Figure 2 shows a theoretical and presumed schematic diagram of intranodal administration of the antigen-pulsed DC vaccine and possible immune response in injected LNs.

However, only about 1% of DCs injected intradermally reach LNs [36], and it is difficult to predict whether they would present antigens to T cells. Even in the method of injecting tumor antigen alone or with adjuvant into LNs, whether antigens are taken up by resident DCs in LNs is also dependent on chance. In contrast, direct injection of antigen-loaded DCs into LNs involves the transfer of mature DCs presenting tumor antigens in the vicinity of T cells. Therefore, the likelihood of direct activation of tumor-antigen-responsive naïve T cells in LNs increases.

Evidence supporting the rationale for intranodal administration of DC vaccines comes from recent reports on intranodal lymphangiography using direct injection of an oil-based contrast agent into inguinal LNs. The original method of lymphangiography was first reported by Rajebi et al. in 2011 [37], followed by Nadolski et al. in 2012 [38], involving US-guided injection of an oily contrast agent into normal inguinal LNs to diagnose and treat areas of lymphatic leakage due to intraoperative lymphatic injury; moreover, its usefulness and safety have recently been reviewed in several reports [39,40]. Anatomically, LNs are connected to lymphatic vessels, and the lymphatic flow from LNs in the thorax, pelvis, and lower extremities is connected to the thoracic duct, which eventually joins the veins above the left clavicle and enters the vascular system. Normal LNs in the groin are large and accessible among the superficial lymphatic vessels. In addition, the contrast agent injected into the groin LNs drains from the efferent lymphatics, migrates from the pelvis to the collecting lymphatics in the abdominal cavity, flows into the thoracic duct, and enters the veins above the left clavicle. On the basis of the theory of lymphangiography, it can be easily inferred that tumor-antigen-responsive lymphocytes activated by DCs administered in normal groin LNs can migrate throughout the body via the upstream lymphatic system, supporting the rationale for LN infusion therapy of DC vaccines.

Theoretically, LNs other than TDLNs can also be sites of activation and proliferation of resident or recruited antigen-responsive T lymphocytes if antigen-presenting DCs are present. The presence of tumor-antigen-presenting mature DCs in normal LNs may possibly facilitate the formation of stronger antitumor immunity than that in the immunosuppressive environment of TDLNs.

## 6. DC Vaccine Intranodal Infusion Therapy

As mentioned above, LNs scattered throughout the body are ideal and perfect small target organs to initiate and amplify tumor immunity. Administration of DCs loaded with tumor antigens into LNs has been attempted since the late 1990s and found to be an excellent method to elicit the efficacy of DC vaccines.

Intranodal administration of DC vaccines in patients with melanoma has been reported since 1998. Sixteen patients with advanced metastatic melanomas were treated with a mature DC vaccine containing class-I-binding tumor-associated antigen peptide with keyhole limpet hemocyanin (KLH) pulsed as helper T cell antigen in normal LNs. A clinical response was shown in 5 of 16 patients [41]. A study comparing the effects of intranodal vaccine administration of immature and mature DCs was reported, and the effects of mature and immature DC vaccines were compared in the same patients [42]. Immunological responses were assessed in 11 patients with stage IV melanoma to intranodal mature DC vaccine pulsed with peptide antigen A (one of tyrosinase, MelanA/MART-1, or MAGE-1) in a normal lymph node. In addition, responses to immature dendritic cell vaccine pulsed with peptide antigen B (another one of tyrosinase, MelanA/MART-1, or MAGE-1) administered in another normal lymph node were evaluated. The results showed that mature DCs are superior to immature DCs in T-cell reactivity to antigens (ELISpot reaction) and the induction of antigen-specific cytotoxic T cells.

Intranodal administration of DCs pulsed with both class I antigen peptides and helper antigen KLH was also studied [43]. It has been demonstrated that intranodal infusion therapy with monocyte-derived mature DC vaccine pulsed with KLH and melanoma-related peptides could induce both class-I-bound-melanoma-related antigen-specific T cell and KLH-specific helper T cell immune responses. Intranodal administration of autologous tumor-lysate-pulsed monocyte-derived mature DC vaccine was reported [44]. An increased immune response of peripheral blood lymphocytes to autologous tumor proteins and reduced tumor size (1 CR, 4 PR) in 5 of 10 patients with T-cell lymphoma was shown.

Intranodal DC vaccine therapies for solid tumors other than melanoma have been reported [45,46]. Eighteen patients with renal cell carcinoma were treated with intranodal infusion of autologous tumor lysate-pulsed DCs, and nine patients, including three with CR, showed tumor reduction. The only adverse event was the cytokine response to concomitant IL-2 and IFN-α. In a report on 26 patients with metastatic colorectal cancer treated with intranodal infusion of autologous tumor-lysate-pulsed monocyte-derived matured DC vaccine for prevention of postoperative recurrence, more than 60% of the patients showed tumor-antigen-specific T-cell responses, and the recurrence-free survival rate was extended to more than 5 years [46].

Intranodal administrations of vaccines pulsing antigen mRNA to DCs instead of antigen peptides were reported [47,48]. Twenty-six patients with stage III melanoma and 19 with stage IV melanoma were treated with intranodal infusion of a monocyte-derived mature DC vaccine pulsed with gp100 and tyrosinase mRNA by electroporation; antigen-specific T-cell immunoreactivity was observed in 17 of the 26 patients in stage III melanoma. Moreover, 15 patients showed tumor reduction; although in stage IV melanoma, antigen-specific immunoreactivity was observed in 11 of 19 patients, tumor reduction was observed in only three patients. Therefore, this study indicated the limited clinical efficacy of vaccine therapy alone in stage IV melanoma [47]. In the analysis of patients treated with gp100 and tyrosinase mRNA as well as co-stimulatory molecules CD40, CD70, and TLR4 mRNA pulsed into monocyte-derived mature dendritic cells and administered into LNs, the clinical efficacy and safety have been reported, with no adverse events observed. Additionally, tyrosinase-specific T lymphocytes were found in lymphocytes of the intradermal reaction site, with a reduction in size in two of eight cases and a mixed reaction in one case [48].

Cases of intranodal DC vaccine therapy for hematological malignancies have also been reported [49]. A study treated nine patients with multiple myeloma with intranodal administration of a monocyte-derived dendritic cell vaccine pulsed with an autologous idiotype protein and KLH and matured with CD40 ligand. The authors concluded that the vaccine was an effective and safe treatment with no adverse events.

As described above, several reports have shown the efficacy of intranodal infusion of monocyte-derived mature DC vaccines, and the results are summarized in Table 1. However, the most common route of DC vaccine administration is intradermal administration. Intradermal administration is simpler and does not require a special US device; it is a manageable technique compared to intranodal administration. Several studies have compared the immune response and efficacy between intranodal administration of DC vaccines and other administration routes, such as intradermal administration, and concluded that the intranodal method was superior [50,51]. For example, the effect of intranodal administration of tumor-lysate-pulsed DC with IV and ID administration of the same DC vaccine on metastatic lesions and tumor-specific T-cell immunoreactivity in a mouse tumor model were compared [50]. IN vaccination was found to have the best antitumor effect and tumor-specific T-cell responses. The immunological effects of a class-I-binding tumor antigen peptide-pulsed DC vaccine in 27 patients with advanced melanoma were evaluated. Moreover, the immunological effects of the DC vaccine against tumor antigens by different routes of administration (intravenous, intradermal, and intranodal) were compared. The results showed that all methods were safe and effective, and the IN group was the best in terms of CD8 T cell response to tumor antigen and delayed-type hypersensitivity (DTH) responses [51].

On the other hand, there have been reports revealing that intradermal administration of a DC vaccine had the same or better immunological effect than did intranodal administration. In a study, 43 patients with advanced melanoma allocated to intradermal (*n* = 21) and intranodal (*n* = 22) groups were treated with a DC vaccine pulsed with melanoma antigens gp100 and tyrosinase peptides together with KLH. The study found that
T-cell immunoreactivity to melanoma antigens was higher in the intradermal group (53%) than that in the intranodal group (16%) [52]. Immune responses to autologous tumors and survival rates between intradermal (*n* = 10) and intranodal (*n* = 21) vaccine groups in advanced melanoma patients were compared, with the intradermal group showing a better response than the intranodal one [53]. In contrast, another study found no significant difference in the immune response to HER2 when a HER2 peptide-pulsed DC vaccine was administered either intratumorally, intranodally, or intratumorally plus intranodally in 54 patients with HER2-positive breast cancer [54].

Thus, the superiority or inferiority of the route of administration of DC vaccines is still controversial, and any route of DC vaccine administration could induce immune responses to varying degrees.

A number of intranodal monocyte-derived matured DC vaccine therapies have been reported, and all of them were proven to be meaningful in terms of safety and immunological and clinical efficacy, as summarized in Table 1. Intranodal DC vaccine therapy has been mostly reported in patients with advanced and metastatic cancers; however, clinical efficacy, such as tumor shrinkage, was observed at most a few tens of a percent, and all reports concluded that it was a safe treatment with very few serious adverse events.

**Table 1 cancers-14-02438-t001:** References.

Reference	Study Type	Tumor	Antigens	Adjuvant/Stimulant	Results	Reference
Nestle FO et al., 1998	Clinical	Melanoma	Tumor Lysate. MelanA/Mart1, Mage1/Mage3	KLH	2 CR and 3 PR in 16 patients	[41]
Lambert LA et al., 2001	Pre-clinical(mice)	Melanoma	Tumor Lysate	( - )	IN superior than ID in Th1 response	[50]
Bedrisian I et al., 2003	Clinical	Melanoma	Peptieds for Mart1, gp100, Tyrosinase	( - )	IN superior than ID in Th1 response	[51]
Jonuleit H et al., 2001	Clinical	Melanoma	MelanA/Mart1, Mage1/Mage3	( - )	in case of matured DC CD4 response 7/8 CD8 response 5/7	[42]
Gilliet M et al., 2003	Clinical	Melanoma	MelanA/Mart1, Mage1/Mage3	KLH	Long lasting CD4T-cell response with TH-1 cytokine response in all 5 patients	[43]
Maier T et al., 2003	Clinical	Lymphoma	Tumor Lysate	KLH	4 PR and 1 CR in 8 patients	[44]
Schwaab TS et al., 2009	Clinical	Renal Cell Cancer	Tumor Lysate	IFN-a2a, IL-2	50% ORR and 3 CR in 18 patients	[45]
Yi Q et al., 2010	Clinical	Multiple Myeloma	Idiotype protein	KLH, CD40	SD in 6 of 9 patients	[49]
Barth RJ et al., 2010	Clinical	Colorectal cancer	Tumor Lysate	KLH, CD40	61% DTH response in 24 patients. 5year recurrence free 63%	[46]
Aarntzen EHJG et al., 2012	Clinical	Melanoma	mRNA for gp100/Tyrosinase	KLH	TAA specific Th1 responsein stage III: median OR 24.1 months	[47]
Bol KF et al., 2015	Clinical	Melanoma	mRNA for gp100/Tyrosinase	CD40,/TLR4 mRNA	1 MR and 2 durable SD in 8 patients	[48]
Morisaki T et al., 2020	Clinical	Ovarian Cancer	Neoantigen peptides	( - )	1 case report: durable SD	[55]
Morisaki T et al., 2021	Clnical	Solid tumors	Neoantigen peptides	( - )	1 CR, 3 PR, and 10 SD in 17 patietns	[56]

## 7. Techniques of Intranodal Administration of Mature DCs

There is a lack of reports detailing how DCs are administered to LNs. To date, most reports state that an expert should perform intranodal DC vaccination under US guidance. Therefore, we used intranodal administration of neoantigen peptide-pulsed DCs under US guidance and reported on this method [55,56].

In our method, inguinal LNs should be determined, using US, to be injected with the vaccine. One hour prior to the start of the procedure, lidocaine tape was applied to the skin. Additionally, a small amount (1 mL) of xylocaine was injected into the skin and fascia surrounding the LNs with a 26G needle syringe under US guidance. Thereafter, a saline solution (0.5 mL of saline solution with DCs suspended in it) was injected into the LNs with a 25G needle under US guidance. The injection target was the border between the cortex and medulla to prevent leakage from LNs. If injected slowly in the direction from the cortex to the medulla, LNs as small as 5 mm in size could be targeted. As a result, LNs increased by 1 to 3 mm in both length and width compared with their size before injection. Export lymph vessels exist from the medulla to the hilum, and in a normal inguinal lymph node, the medulla is caudal to cephalic and relatively long. Therefore, the injection target was from caudal to cephalad, and the needle tip was fixed at the border of the LN cortex and medulla and injected slowly with a 25G Catheline Syringe needle. Using this method, we did not observe any changes in LN structure even after several (up to six) injections of the vaccine.

There are two puncture methods under US guidance: parallel and intersecting or cross. The parallel method involves puncturing the target in parallel from the short part of the probe, while the cross method involves puncturing from the side of the long part of the probe. The parallel method has an advantage: the echo of the needle can be seen. However, even if only the echo of the tip of the needle can be seen in the cross method, the target is wider and can be punctured at a shorter distance. Another difference is that the parallel technique confirms the size and depth of the LN and the direction from the medulla to the hilum, while the cross technique is more reliable for puncturing at the border between the cortex (low-density echo) and medulla (high-density echo) of the LN. Therefore, we recommend the cross technique for intranodal injection after confirming the LN size and direction of the target using parallel techniques.

Figure 3 shows changes in the diameter of LNs before and after the DC vaccine, categorized by the diameter of the original LN. In all cases, 5 mm or even smaller LNs were successfully injected with the vaccine. Compared to their size before treatment, the diameter of LNs increased by more than 3–4 mm after three treatments, indicating LN expansion. Increases in LNs after intranodal DC vaccination may involve activation of FRCs by injected mature DCs [13].

In gynecological tumor surgery patients with lymphedema of the lower limbs due to pelvic LNs dissection, inguinal LN injection should be performed with caution. In cases where lymphedema worsens with LN infusion, switching to intra-axillary LN administration is often necessary.

## 8. Intranodal Neoantigen Peptide-Pulsed DC Vaccine Therapy

One of the most important factors for DC vaccines is the tumor antigen pulsed to DCs. Among the tumor antigens, neoantigens are the most tumor specific and can induce an antitumor response in T cells [57]. Peptides with amino acid substitutions resulting from genetic mutations in cancer cells can be potent cancer rejection antigens because they evade central immune tolerance in the thymus. Recently, neoantigens based on the expression of genes with amino acid substitutions caused by mutations in cancer and the predicted affinity of the predicted peptides to HLA class I have been attracting attention. HLA class-I-affinity neoantigen peptides could be used as materials for tailor-made vaccines in individual patients [58]. Phase I/Ib clinical trials have reported the safety and efficacy of neoantigen vaccines, and many clinical trials have been initiated as single or combined immunotherapy [58,59]. There are various forms of neoantigens, varying from vaccines with neoantigen peptides converted to mRNA and vaccines with neoantigen long peptides alone or plus immune adjuvants, to vaccines with neoantigen peptides pulsed into DCs, such as the ones we are currently using. Further studies are needed to confirm the method used to determine the efficacy, advantages, and disadvantages of these vaccines.

We have established one cancer vaccine method, one that involves synthesizing predicted neoantigen epitope peptides based on genetic analysis of an individual patient’s tumor, pulsing it into dendritic cells and administering it to the patient’s lymph nodes [55,56]. This method was started for cancer immunotherapy as a type III regenerative medicine under the Act to Ensure the Safety of Regenerative Medicine in Japan. The results have been reported in papers that analyzed the clinical and immunological efficacy [56]. We showed that the efficacy of the vaccine was correlated with an increase in the number of neoantigen-reactive T-cell responses determined by IFN-γ ELISpot analysis, resulting in an extremely safe immunotherapy with almost no adverse events.

## 9. Conclusions and Perspective

Structurally, LNs can be considered complete immune organs distributed throughout the body. LNs can be targeted directly in cancer immunotherapy and are expected to be a delivery device for drugs and vaccines. Intranodal-tumor-antigen-pulsed DC vaccines are safe and effective. However, novel therapeutic approaches need to be developed: the direct access to LNs requires technique and skill under US guidance, and advancement of the US guidance function is also necessary for a reliable intranodal injection procedure. Clinical trials are required to validate the US-guided intranodal injection of drugs, vaccines, and cellular products.

## Figures and Tables

**Figure 1 cancers-14-02438-f001:**
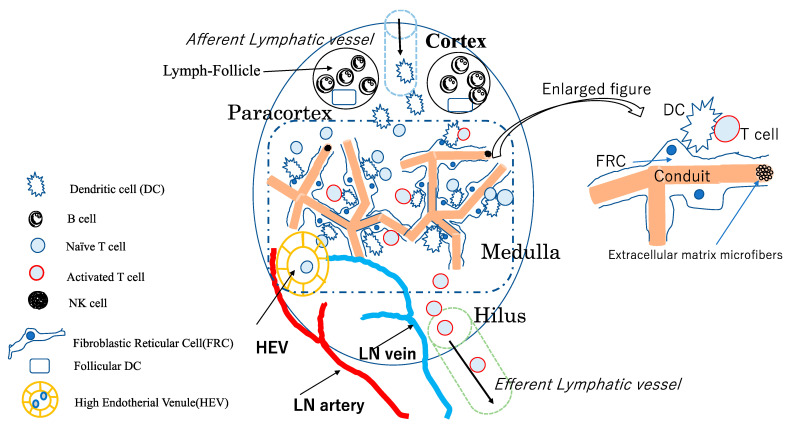
Schematic diagram of an immune-cell-based structure of a single LN. Mature DCs expressing CCR7 that take up antigens in surrounding tissues. They flow into the lymph node according to a concentration gradient of chemokines, such as CCL21, produced by the lymphatic epithelium, through the afferent lymphatic vessel that enters the lymph node capsule. In the cortex under the lymph node capsule, there are lymph follicles that accumulate B-lymphocytes and FDCs that produce chemokines accumulating B-lymphocytes. Between the cortex and medulla, there is a paracortex, which mainly contains antigen-presenting cells and T cells. Antigen-responsive naïve T cells, which receive antigen stimulation and co-stimulation from DCs, are activated and become effector T cells, which migrate to the medulla and enter the efferent lymphatic vessel. Between the arteries and veins of the lymph nodes, there are special blood vessels (HEVs) in the paracortex and medulla that express surface antigens such as peripheral node addressin (PNAd) and ICAM-I/II for lymphocyte rolling and adhering to endothelial cells. They produce chemokines that allow lymphocytes to enter the lymph nodes by extravasation.

**Figure 2 cancers-14-02438-f002:**
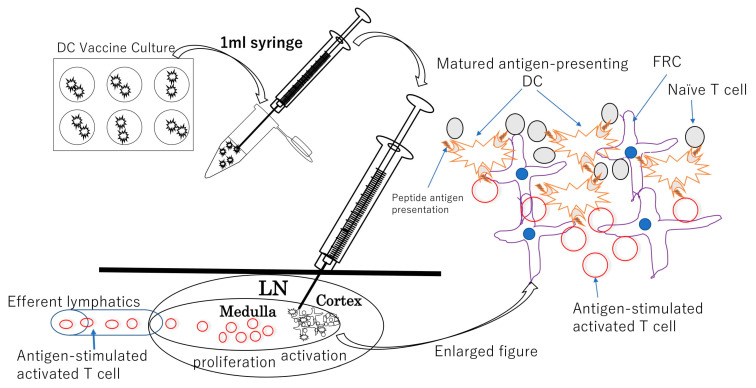
Schematic diagram of intanodal DC vaccine injection under US guidance and possible immunological response in the LN. Antigen-pulsed mature dendritic cells are punctured to target the cortex and medulla border under US guidance. Approximately 0.5 mL of saline containing DC vaccine is injected into the lymph nodes with relatively little resistance. The paracorotex and medulla contain a large number of resident T lymphocytes and lymphocytes recruited from blood via HEV and antigen-responsive T lymphocytes that react with antigen-presenting DCs, where they are activated and co-stimulated by antigen-presenting DCs and proliferate. Antigen-stimulated effector and memory T cells migrate to the vasculature via efferent lymphatics, from where they are thought to spread throughout the body.

**Figure 3 cancers-14-02438-f003:**
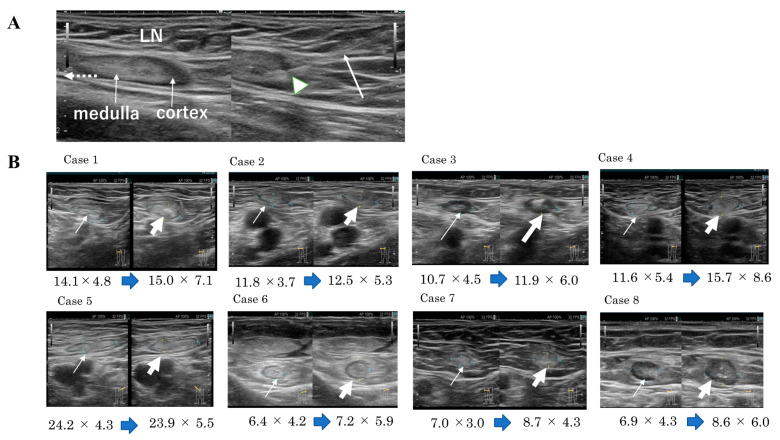
Intranodal injection of dendritic cell vaccine (**A**) An ultrasound-guided inguinal lymph node puncture. Left, before puncture; right, during puncture. A 25-G puncture needle (white arrow) was used under ultrasound guidance to puncture the inguinal lymph node. The tip of the needle is located at the border between the cortex and medulla of the lymph node. (**B**) Changes in lymph node diameter before and after intranodal injection of dendritic cell vaccine. Changes in lymph node diameters (mm) before and after intranodal injection of dendritic cell vaccine in eight patients are shown. In each case, ultrasound images of the lymph nodes before and after injection are shown on the left and right, respectively (thin arrows indicate pre-injection lymph nodes and thick arrows indicate post-injection lymph nodes). The width and height of each node increased by 1~3 mm after injection.

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
