# Peer review of "Lymph Nodes as Anti-Tumor Immunotherapeutic Tools: Intranodal-Tumor-Specific Antigen-Pulsed Dendritic Cell Vaccine Immunotherapy"

_cancers, 2022, doi:10.3390/cancers14102438_

Round 1
Reviewer 1 Report
Lines 59-61 should be deleted
Line 63 sentence to be riformulate (it is too short and wo references)
Figures should be more complex and enriched of more information. As it is presented has not any add value.
Abbreviations have to be revised
Author Response
To Reviewer 1
Thank you for your valuable comments. Corrections and additional statements have been made to all comments. Please see the corrections below, one by one, and Figures has also been changed to a more informative version in accordance with your comments. We have also added references. Please see below for a description of each of them.
your comments (red)
our response (black)
Lines 59-61 should be deleted
We deleted lines 59-61.
Line 63 sentence to be riformulate (it is too short and wo references)
We reformulated the sentences (line63-line65) as follows.
To create cancer vaccine, it is important to consider what to use as the antigen, how to deliver that antigen to the body's immune cells, and the route of administration of the vaccine (ref). One method of cancer vaccine therapy is to pulse cancer-specific antigens to patients’ dendritic cells outside the body, which are then administered into the lymph nodes.
Figures should be more complex and enriched of more information. As it is presented has not any add value.
We elaborated on the diagram according to the reviewer’s comments.
Abbreviations have to be revised
lymph nodes (LNs), dendritic cells (DCs), ultrasound (US), tumor-draining LNs (TDLNs), immune checkpoint inhibitors (ICIs), fibroblastic reticular cells (FRCs), LN stromal cells (LNSCs), follicular DCs (FDCs), high endothelial venule (HEV), vascular endothelial growth factor (VEGF), tumor microenvironment (TME), keyhole limpet hemocyanin (KLH)
Revised Figures are attached.

Reviewer 2 Report
The manuscript reviews the field of intranodal injection of cultured and antigen pulsed DCs as cancer treatment. overall the paper is relevant and reasonably well written. There are some issues with unclear phrasing and a general tendency to make broad statements about issues in which the published data is either incomplete or there are disagreeing publications. The paper would be improved by a bit more focus on the relevant issue and less background on lymph node biology. Further improvement could come from an expanded section on what the challenges are in the field and why despite an impressive list of good responses in trials, this is not moving faster into clinical usages. what are the unanswered questions holding this back?
specific issues of communication: I do not attempt to list all less than optimal phrasing but highlighted many and make a list of some examples here:
the following lines have clumsy or unclear phrasing
20, 37-8, 121-2, 144-5, 214-5, 420, 429-30
125-6: migration is not "by HEV producing CCL21)
line 100-1, states T cells eliminate antigens, they eliminate other cells that carry antigen
section on LN structure is quite extensive and could be shortened without hurting the paper, like 109-112
146 lists T cells as APCs
150 calls addressins surface antigens, in the context of this paper that is confusing
158-9 TME is not the site of elimination of immune cells against the tumor and TME is not well understood
163 "they" is not clear as to what it refers to
169 that all DC are "in an immature state in tumor-area LNs" is overstatement, although most are immature
174: stating that "LN in tumor vacinity are immunosuppressive" is again overstatement, needs modification with a word like "most"
184 response is not "to TDLN"
221-2 needs explanation, it is not obvious that direct injection into LN of DC is more likely to activate T cells
229: DCs "are punctured" is not correct
239-240 is not a complete sentence
260-1: it is not clear that intranodal injection of DCs supports broad distribution of DC, this is conjecture and not supported by references
263-7 paragraph is unclear,
271: overstated, it does not "ensure" efficacy, that implies every patient responds
276: 5 patients responded out of how many?
434: dont seem to fit the definition of retrospective studies which implies looking back at studies not done to specifically test the hypothesis of interest
444: what advancement of US guidance function is needed?
Author Response
To Reviewer 2
Thank you very much for your many valuable comments on the entire paper and your suggestions for further improvement of the paper. Corrections and additional statements have been made to all comments. Please find the corrections below, one by one.
Intranodal therapy of vaccines and drugs is looking more and more promising, but unlike other routes of administration, it is difficult to standardize and requires skill in the technique, which I think is the reason why it has not progressed well.
Therefore, we believe that our review of intranodal administration therapy will undoubtedly have an impact on the reader's research and clinical practice in this area.
Your comments (red)
Our response or correction (black)
Comments for 20, 37-8, 121-2, 144-5, 214-5, 420, 429-30
20
We rewrote “the line 20 sentence” as follows.
In this review, we highlight current topics on lymph node structure important in anti-tumor immunotherapy and intranodal antigen-presenting mature dendritic cell vaccine therapy.
37-38
Thus, combined resection of regional LNs with tumor assist in preventing distant metastasis and local recurrence of tumor.
121-122
The influx immune cells such as T cells into LNs occurs mainly via the high endothelial venule (HEV),
125-6: migration is not "by HEV producing CCL21)
Binding of lymphoid cells to HEVc is also supported by an arrest chemokine such as HEV-producing CCL21.
214-215
Although ICIs can restore T cell exhaustion, but there are issues to be resolved, such as priming and activation of T cells by tumor antigen-presenting cells and their accumulation in tumors.
420
We change “they” as HLA class-I-affinity neoantigen peptides
429-430
We rewrote the sentence as follows.
We have established one cancer vaccine method, which involves synthesizing predicted neoantigen epitope peptides based on genetic analysis of an individual patient's tumor, pulsing it into dendritic cells, and administering it to the patient's lymph nodes.
line 100-1, states T cells eliminate antigens, they eliminate other cells that carry antigen
We corrected and rewrote as follows.
They spread throughout the body, respond directly or indirectly to antigens, and are involved in elimination of other cells that carry pathogenic and tumor-associated antigens
section on LN structure is quite extensive and could be shortened without hurting the paper, like 109-112
Because we believe the 109-112 sentences are relatively important, we do not want to cut it.
Let me leave that sentence as it is.
146 lists T cells as APCs
We rewrote the sentence (145-146) as follows.
Between cortex and medulla there is an paracortex, which mainly contains antigen presenting cells and T cells.
We also deleted the sentence (152-154) to shorten the section.Instead, we have included the following sentence as a description of the FRC-generated comnduit system, which we additionally drew in the upper right corner of Figure 1.
150 calls addressins surface antigens, in the context of this paper that is confusing
We deleted the sentence containing “ addressins” (150-151) because there is a description in the text that contain “addressins”
158-9 TME is not the site of elimination of immune cells against the tumor and TME is not well understood
We corrected it as follows.
Which is the site of direct activation of immune cells against tumor,
163 "they" is not clear as to what it refers to
They ⇨ TDLNs
169 that all DC are "in an immature state in tumor-area LNs" is overstatement, although most are immature
We corrected it as follows,
DCs, which induce tumor antigen presentation to and activation of T lymphocytes, are mostly in an immature states in tumor-area LNs and
174: stating that "LN in tumor vacinity are immunos
uppressive" is again overstatement, needs modification with a word like "most"
We corrected it as follows,
Most LNs in tumor vicinity may be highly immunosuppressive.
184 response is not "to TDLN"
We corrected it as follows,
Anti-tumor immune response in TDLN
221-2 needs explanation, it is not obvious that direct injection into LN of DC is more likely to activate T cells
We rewrote the sentence adding one sentence for explanation as follows.
As mentioned above, not only do many T lymphocytes stay in the LNs, but there is also a constant influx of T lymphocytes via HEVs, so as long as antigen-presenting mature dendritic cells are present, tumor antigen-reactive T lymphocytes are likely to encounter them, become activated, and proliferate. Hence, theoretically, direct injection of mature DCs presenting tumor antigens to LNs is more likely to activate antigen-responsive T lymphocytes.
229: DCs "are punctured" is not correct
We rewrote it as follows,
Antigen-pulsed mature dendritic cell vaccines are injected to target the cortical-medullary interface of LNs.
239-240 is not a complete sentence
We rewrote the sentence as follows,
Even in the method of injecting tumor antigen alone or with adjuvant into LNs, whether antigens are taken up by resident DCs in LNs is also dependent on chance.
260-1: it is not clear that intranodal injection of DCs supports broad distribution of DC, this is conjecture and not supported by reference
Lymphocytes that are activated and proliferated by antigen stimulation in the lymph nodes are thought to flow from the export lymph vessels to the collecting lymph vessels and thoracic ducts, and then join the subclavian vein for distribution throughout the body.
263-7 paragraph is unclear,
We rewrote the 263-266 sentence as follows,
It can be inferred that tumor antigen-presenting dendritic cells are more potent stimulators of antigen-reactive T cells in normal LNs than in TDLNs in an immunosuppressive environment.
271: overstated, it does not "ensure" efficacy, that implies every patient responds
We rewrote the sentence as follows.
Administration of DCs loaded with tumor antigens into LNs has been attempted since the late 1990s and found to be an excellent method to elicit the efficacy of DC vaccines.
276: 5 patients responded out of haw many?
Rewrote it as follows,
In five of 16 patients
434: dont seem to fit the definition of retrospective studies which implies looking back at studies not done to specifically test the hypothesis of interest
We deleted “ retrospective studies” and added “ papers which analyzed the clinical and immunological efficacy
444: what advancement of US guidance function is needed?
We rewrote it as follows.
[Advancement of the US guidance function such as probe accessory to guide needle tip position precisely]
Revised figures are attached.

Reviewer 3 Report
The paper of Morisaki et al. summarizes advances in intranodal dendritic cell-based vaccine immunotherapies. This is an original and well written paper. I have no serious concerns except some important information on the lymph node architecture that is missing in the introductory part of the manuscript. What I think is important and should be added to the text is description of the tubular-like structures formed by reticular fibroblasts and bundles of the reticular collagen type III fibers that are believed to play a crucial role in specific distribution of lymph along the cortical and paracortical area. I’d like also to report that the references appear not to conform the style of the journal.
Author Response
To Reviewer 3
The paper of Morisaki et al. summarizes advances in intranodal dendritic cell-based vaccine immunotherapies. This is an original and well written paper. I have no serious concerns except some important information on the lymph node architecture that is missing in the introductory part of the manuscript. What I think is important and should be added to the text is description of the tubular-like structures formed by reticular fibroblasts and bundles of the reticular collagen type III fibers that are believed to play a crucial role in specific distribution of lymph along the cortical and paracortical area. I’d like also to report that the references appear not to conform the style of the journal.
Thank you for your valuable comments on the structure of the lymph nodes, I have made an additional description of the conduit system, a tubular-like structure by FRC, in Figure 1 and Text.
We have followed the reviewer’s comments and added a diagram of FRC-producing tube-like structure (conduit system) in LNs to Figure 1 and the legends, and the text as follows.
(Figure 3)
An enlarged figure of tubular like collagen (Conduit System) by FRC is added in the upper right corner of Figure 1. The flow of lymphocytes from HEV to Paracortex, to Medulla, to encounter DCs, and to flow out of the export lymph vessels after activation is shown by curved arrows to illustrate the flow of lymphocytes in the lymph nodes. In addition, the picture of lymph node infusion in Figure 2 was moved to Figure 3 A to better understand intranodal injection method and the results (LN enlargement).
(Text)
One of the important functions of the FRC is the formation of the conduit system, which is microfibers of the extracellular matrix through which lymph fluid containing antigens and inflammatory mediators flows, contributing to the control of immune cells in the lymph nodes.
We also added a review of FRC's conduit system.
Acton SE; Onder L.: , Novkovic M,;, Martinez VG,; Ludewig B. Communication, construction, and fluid control: lymphoid organ fibroblastic reticular cell and conduit networks.
Trends Immunol. 2021, 42, 782-794. doi: 10.1016/j.it.2021.07.003. PMID: 34362676
Revised Figures are attached.

Round 2
Reviewer 3 Report
The authors corrected the manuscript in line with the comments.
Author Response
Thank you very much for your comments.
The English editing was done by Editage Co Ltd.
We attached the final version of our manuscript.
